# Improving the Training of GANs with Limited Data via Dual Adaptive Noise Injection

## ABSTRACT

Recently, many studies have highlighted that training Generative Adversarial Networks (GANs) with limited data suffers from the overfitting of the discriminator ($D$). Existing studies mitigate the overfitting of $D$ by employing data augmentation, model regularization, or pre-trained models. Despite the success of existing methods in training GANs with limited data, noise injection is another plausible, complementary, yet not well-explored approach to alleviate the overfitting of $D$ issue. In this paper, we propose a simple yet effective method called Dual Adaptive Noise Injection (DANI), to further improve the training of GANs with limited data. Specifically, DANI consists of two adaptive strategies: adaptive injection probability and adaptive noise strength. For the adaptive injection probability, Gaussian noise is injected into both real and fake images for generator ($G$) and $D$ with a probability $p$, respectively, where the probability $p$ is controlled by the overfitting degree of $D$. For the adaptive noise strength, the Gaussian noise is produced by applying the adaptive forward diffusion process to both real and fake images, respectively. As a result, DANI can effectively increase the overlap between the distributions of real and fake data during training, thus alleviating the overfitting of $D$ issue. Extensive experiments on several commonly-used datasets with both StyleGAN2 and FastGAN backbones demonstrate that DANI can further improve the training of GANs with limited data and achieve state-of-the-art results compared with other methods.

## CCS CONCEPTS

• **Computing methodologies** → **Image representations**; *Computer vision representations*; Neural networks.

## KEYWORDS

GANs, Limited Data, Dual Adaptive Noise Injection.

**ACM Reference Format:**
Anonymous MM submission. 2024. Improving the Training of GANs with Limited Data via Dual Adaptive Noise Injection. In *Proceedings of the 32nd ACM International Conference on Multimedia (MM '24)*. ACM, New York, NY, USA, 10 pages. https://doi.org/XXXXXXX.XXXXXXX

---

*

## 1 INTRODUCTION

In recent years, Generative Adversarial Networks (GANs) [14] have achieved great success in generating contents, e.g., images [24, 27, 28, 59], videos [8, 17, 41], text [18, 53] and audio [29], for social media. These generated contents can be applied in various multimedia applications, such as talking face [55, 60]. However, the success of GANs relies on the availability of a large amount of data. Collecting and cleaning these large datasets can be expensive, time-consuming, and even infeasible. Consequently, training GANs with limited data has received great attention.

Recently, several approaches [7, 26, 65] have demonstrated that training GANs with limited data suffers from the overfitting of the discriminator ($D$), i.e., $D$ becomes overly confident in distinguishing between real and fake data. To address this, existing methods employ various strategies such as data augmentation [7, 9, 23, 26, 65], model regularization [12, 33, 51], or pre-trained models [30, 45]. Despite the success of existing methods, noise injection [1, 4] is another plausible, complementary, yet not well-explored approach to alleviate the overfitting of $D$. Recent noise injection methods in GANs [2, 13, 22, 43, 47] have already shown their effectiveness in improving the training of GANs with a large amount of data. However, as stated in ADA [26], directly applying noise injection to GANs with limited data suffers from the leaking problem [26, 63, 64, 66], i.e., "noise augmentation leads to noisy results, even if there is none in the dataset", which can highly influence the performance of GANs training under limited data settings.

In this paper, we propose a novel noise injection method for GANs with limited data, called Dual Adaptive Noise Injection (DANI). Specifically, DANI consists of two adaptive strategies, i.e., adaptive injection probability and adaptive noise strength. For the adaptive injection probability, we inject Gaussian noise into both real and fake images for $G$ and $D$ with a probability $p$, where the $p$ is controlled by the overfitting degree of $D$ adaptively. For the adaptive noise strength, we apply the adaptive forward diffusion process [20] to both real and fake images, respectively, to produce the Gaussian noise. Consequently, both adaptive strategies in DANI can effectively prevent the leaking problem caused by the noise injection. Furthermore, DANI can effectively alleviate the overfitting of $D$ problem, i.e., $D$ becomes overly confident in distinguishing the real and fake data. Specifically, we prove that DANI effectively increases the overlap between the supports of real and fake data distributions in GANs during training (see Theorem 3 in §3.2), as illustrated in Figure 1. Then, based on the conclusion in ADA [26], the increased overlap between the supports of real and fake data distributions provided by DANI strongly indicates that DANI mitigates $D$ becoming overly confident in distinguishing real from fake data. This demonstrates that DANI alleviates the overfitting of $D$ problem.

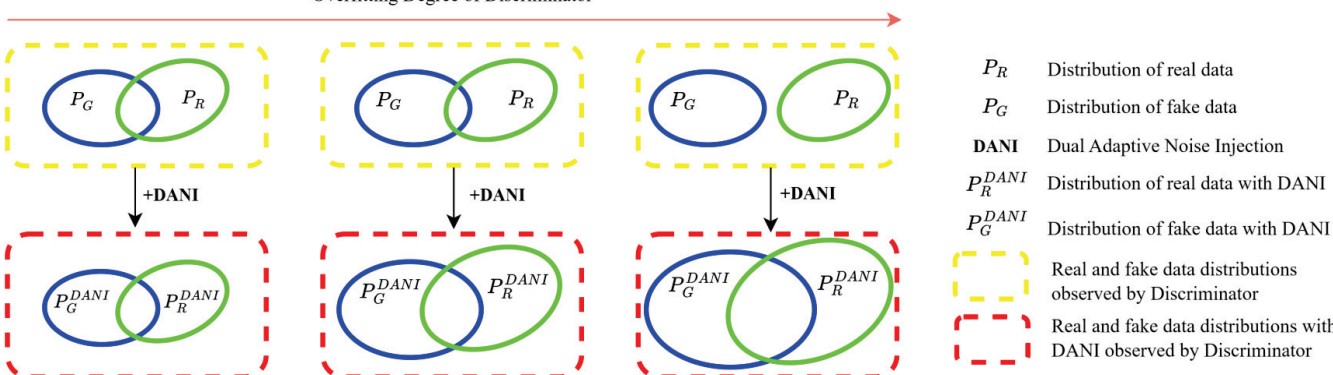

**Figure 1: Schematic overview illustrating how DANI can benefit the training of GANs with limited data. Top: The real and fake data distributions observed by $D$ during training. As the overfitting of $D$ increases, $D$ becomes increasingly confident in distinguishing between real and fake data, leading to a decrease in the overlap between the supports of real and fake data distributions observed by $D$. Finally, when $D$ becomes overly confident, i.e., when $D$ reaches optimality as pointed out in [2, 14], the overlapping parts disappear or can be ignored. Bottom: The real and fake data distributions with DANI observed by $D$ during training. Applying DANI for both real and fake samples can effectively provide more overlapping parts between the supports of real and fake distributions during training, thus resulting in better performance [2, 54].**

To sum up, the main contributions of this paper are as follows:

(1) We propose a novel method called Dual Adaptive Noise Injection (DANI) for training GANs with limited data. DANI can effectively alleviate the overfitting of $D$ and avoid the leaking problem.
(2) We provide the theoretical analysis of DANI for training GANs with limited data, proving its convergence and rationality on both StyleGAN2 [28] and FastGAN [33] backbones. Furthermore, we prove that applying DANI to GANs can provide more overlapping parts between the supports of the real and fake data distributions.
(3) Extensive experiments on several commonly used datasets demonstrate that the proposed DANI can further improve the training of GANs with limited data and achieve state-of-the-art performance compared with other methods. Additionally, DANI achieves these benefits with only a negligible increase in computational cost.

## 2 PRELIMINARIES

### 2.1 Generative Adversarial Networks (GANs)

Generative adversarial networks (GANs) [14] is a form of generative models [40, 52] in which a game is played between two players: A generator ($G$) and a discriminator ($D$). Specifically, $G$ aims to produce realistic-looking samples with some given noise $z$ to deceive $D$, while $D$ aims to distinguish whether the input sample is from the generator's output or real data. The objective function of GANs can be formulated as follows:

$$\min_G \max_D V(D, G) = \mathbb{E}_{x \sim P_R}[\log D(x)] \\ + \mathbb{E}_{x \sim P_G}[\log(1 - D(x))]. \quad (1)$$

The parameters of $G$ and $D$ are updated iteratively with gradient descent methods. GANs are known to suffer from training instability, yielding poor quality and low diversity of generated images. To stabilize GANs training as well as improve the quality and diversity of generated images, various approaches have been proposed, focusing on more sophisticated network architectures [6, 38, 39, 42, 58, 61], more stable objective functions [3, 15, 16, 36, 44], and better training strategies [10, 24, 31, 34, 35, 56, 62] to achieve photorealistic results.

### 2.2 Training GANs with Limited Data

**Training GANs with limited data using data augmentation.** Recently, data augmentation (DA) has played an important role in improving performance when training GANs with limited data. Many studies [7, 26, 50, 65] apply DA to both real and fake images for $D$ and $G$ to guide the discriminator avoiding overfitting. The most popular methods are Diff-Augment [65], ADA [26], and Diffusion-GAN [54]. Diff-Augment applies the DAs to both real and fake images for the $D$ and the $G$ without manipulating the target distribution. ADA is similar to Diff-Augment, while it further devises an adaptive approach that controls the strength of data augmentations. Diffusion-GAN applies the adaptive forward diffusion process [20] as DA to both real and fake images.

**Training GANs with limited data using model regularization.** In recent years, several approaches have proposed novel model regularization methods to alleviate the overfitting of $D$, thus improving the training of GANs with limited data. The most significant methods include $R_{LC}$ [51] and DigGAN [12]. Specifically, $R_{LC}$ applies a regularized objective function for the discriminator to improve the training of GANs with limited data. DigGAN applies discriminator gap regularization to alleviate the overfitting of $D$.

**Training GANs with limited data using a pre-trained network.** Recently, employing pre-trained network has been shown to

**Figure 2: An overview of applying DANI to GANs with limited data. DANI consists of the adaptive injection probability strategy and the adaptive noise strength strategy. For the adaptive injection probability strategy, the Gaussian noise is adaptively injected into both the real and fake images for $G$ and $D$ with a probability $p$, where $p$ is controlled by the overfitting degree $\eta$. For the adaptive noise strength strategy, the Gaussian noise is produced by applying an adaptive forward diffusion process to both the real and fake images for $D$ and $G$, controlled by Eq.(7). In DANI, the parameters of $D$ are optimized using noise-injected real and fake samples (all paths), and the parameters of $G$ are optimized using noise-injected fake samples (red paths).**

significantly improve GANs training when using limited data. Many approaches [30, 45] apply a pre-trained network to the discriminator to extract the features from both the real and fake images to improve the training of GANs with limited data. One of the pioneering methods is Projected GAN. Specifically, Projected GAN applies a pre-trained network, i.e., EfficientNet-lite1 [48], to both the real and fake images to extract projected features in image space. Then, Projected GAN applies a multi-scale discriminator architecture which can better utilize deeper layers of the pre-trained network. As a result, Projected GAN can effectively utilize the obtained projected features to benefit the training of GANs with limited data. The loss function of the Projected GAN can be formulated as:

$$\min_{G} \max_{\{D_l\}} \sum_{l \in \mathcal{L}} \{ \mathbb{E}_{x \sim P_R}[\log D_l(P_l(x))] \\ + \mathbb{E}_{x \sim P_G}[\log(1 - D_l(P_l(x)))]\}, \quad (2)$$

where $\{P_l\}$ is the set of feature projectors which map real and generated images to the discriminator's input space, $\{D_l\}$ is a set of independent discriminators operating on different feature projections and $\mathcal{L} = \{1, ..., n\}$ is index set for the different projectors.

## 2.3 Diffusion-based Generative Model

In recent years, Diffusion-based generative models [11, 20, 25] have shown their superiority in image-generation tasks. Diffusion-based generative models assume that $p_\theta(x_0) := \int p_\theta(x_{0:T}) dx_{1:T}$, where $x_1, ..., x_T$ are latent variables of the same dimensionality as the data $x_0 \sim p(x_0)$. There is a forward diffusion chain that gradually adds noise to the data $x_0 \sim q(x_0)$ with pre-defined variance schedule $\beta_t$ and variance $\sigma^2$ in $T$ steps:

$$q(x_{1:T}|x_0) := \prod_{t=1}^{T} q(x_t|x_{t-1}), \quad (3)$$

$$q(x_t|x_{t-1}) := N(x_t; \sqrt{1-\beta_t}, \beta_t\sigma^2 I).$$

A notable property is that $x_t$ at an arbitrary time-step $t$ can be sampled in closed form as:

$$q(x_t|x_0) := N(x_t; \sqrt{\bar{\alpha}_t}x_0, (1 - \bar{\alpha}_t)\sigma^2 I), \quad (4)$$

where $\alpha_t := 1 - \beta_t, \bar{\alpha}_t := \prod_1^t \alpha_s$. A variational lower bound [5] is then used to optimize the reverse diffusion chain as:

$$p_\theta(x_{0:T}) := N(x_T; 0, \sigma^2 I) \prod_{t=1}^{T} p_\theta(x_{t-1}|x_t). \quad$$

## 2.4 Improving GANs Training via Noise Injection

In recent years, several studies have injected noise [2, 22, 43, 47] into the discriminator's input to improve the training of GANs. Specifically, they add random Gaussian noise to the discriminator's input in GANs to achieve better performance. The technique can be formulated as follows:

$$c' = c + \lambda X, \qquad X \sim N\{0, \sigma^2\}, \quad (5)$$

where $\lambda$ is the parameter to scale the noise, $c$ represents the original data, and $c'$ represents the data obtained after noise injection.

## 3 METHODOLOGY

### 3.1 Dual Adaptive Noise Injection (DANI)

Despite the success of existing methods, noise injection is another plausible, complementary, yet not well-explored approach to alleviate the overfitting of $D$ issue. To effectively utilize noise injection to improve GANs training with limited data, in this paper, we propose a novel noise injection method for training GANs with limited data, called Dual Adaptive Noise Injection (DANI). The overview of applying Dual Adaptive Noise Injection (DANI) to the GANs with limited data is shown in Figure 2. DANI consists of two adaptive strategies, i.e., adaptive injection probability and adaptive noise strength.

For the adaptive injection probability, we perform the noise injection based on a form of probability, $p \in [0, 1)$ for each real and fake sample. It is intuitive to let $p$ be adjusted adaptively based on

the degree of overfitting without manual tuning regardless of data scales and properties. To achieve this goal, following ADA [26] and APA [23], we apply an overfitting heuristic $\eta$ that quantifies the degree of $D$'s overfitting as follows:

$$\eta = \mathbb{E}(\text{sign}(D_{real})), \tag{6}$$

where sign() indicates the sign function that returns +1 for a non-negative input; −1, otherwise; $D_{real}$ is the discriminator output on the real training data. Then, we set a threshold value $d_{\text{target}}$ and follow the same step as in ADA [26] and APA [23] for using $\eta$ to adjust $p$. Specifically, we initialize $p$ to zero and adjust its value once every four minibatches based on the chosen overfitting heuristic $\eta$. If $\eta$ signifies too much/little overfitting regarding $d_{\text{target}}$ (i.e., larger/smaller than $d_{\text{target}}$), $p$ will be increased/decreased by one fixed-step. Using this step size, $p$ can increase from zero to one in 500k images shown to $D$. We adjust $p$ once every four iterations and clamp $p$ from below to zero after each adjustment. In this way, the noise injection probability can be adaptively controlled based on the degree of overfitting.

For the adaptive noise strength, we follow Diffusion-GAN, aiming for the strength of the injected noise to be effectively controlled by the forward diffusion process. To this end, we use the same $\eta$ and $d_{\text{target}}$ from the adaptive injection probability strategy to control the time-step $t$ in the forward diffusion process as follows:

$$t = t + \text{sign}\left(\eta - d_{\text{target}}\right) \times C, \tag{7}$$

where $C$ is a constant. We update $t$ every four minibatches based on the chosen overfitting heuristic $\eta$. In this case, the strength of the noise in DANI can be controlled with an adaptive forward diffusion process.

To sum up, DANI can be formulated as:

$$\hat{x} = x + pA_t(x), \tag{8}$$

where $A_t$ is the adaptive forward diffusion process, $x$ represents the original data, and $\hat{x}$ represents the data obtained after noise injection. Following Diffusion-GAN [54], we select the priority distribution for the forward diffusion process and the time step $t$ is controlled adaptively following Eq.(7). According to [54], the forward diffusion process variances $\beta_t$ in Eq.(3) are increased linearly from $\beta_1 = 10^{-4}$ to $\beta_T = 0.02$.

## 3.2 Theoretical Analysis

Let $P_R$ be the distribution of real data and $P_G$ be the distribution of generated data. For the sample $x$, $D(x)$ represents the estimated probability of sample $x$ being real or fake. To examine the rationality of DANI, we analyze it in a non-parametric setting, where a model is represented with infinite capacity by exploring its convergence in the space of probability density functions. Ideally, the estimated probability distribution $P_G$ should perfectly model the distribution $P_R$ without bias if given enough capability and training time.

Since the probability $p$ is adaptively adjusted, to facilitate this theoretical analysis, we assume that $\alpha$ is the expected strength, which approximates the effect of dynamic adjustment of distribution during the entire training procedure. Since $p \in [0, 1)$, we have $0 \leq \alpha < p_{max} < 1$, where $p_{max}$ is the maximum probability throughout training. Based on Eq.(1), the saturating loss function

of DANI in GANs is shown as follows:

$$\min_G \max_D \quad \mathbb{E}_{x \sim P_R}[\log D(x + \alpha A_t(x))] \\ + \mathbb{E}_{x \sim P_G}[\log(1 - D(x + \alpha A_t(x)))], \tag{9}$$

where $A_t$ is the adaptive forward diffusion process, where the forward diffusion process is based on Eq.(3) and the time step $t$ is controlled adaptively following Eq.(7). Ideally, for $\forall x$, $A_t(x) \sim N\{0, \sigma^2\}$, it demonstrates that $A_t(x)$ can be directly regarded as Gaussian noise with different strengths during training. As a result, DANI in the GANs can be theoretically regarded as adding a set of Gaussian noise during training. In this case, we follow [22] to analyze the convergence of Eq. (9).

For $\forall x$, let $\varepsilon = \alpha A_t(x)$ represent the random Gaussian noise. Then, Eq.(9) can be formulated as:

$$\min_G \max_D \sum_{P_\varepsilon \in S} \{\mathbb{E}_{\varepsilon \sim P_\varepsilon}[\mathbb{E}_{x \sim P_R}[\log D(x + \varepsilon)]] \\ + \mathbb{E}_{\varepsilon \sim P_\varepsilon}[\mathbb{E}_{x \sim P_G}[\log(1 - D(x + \varepsilon))]]\}, \tag{10}$$

according to [22], we introduce a set $S$ of probability density functions in Eq.(10), where $\varepsilon \sim P_\varepsilon \in S$. Then, based on [22, 45], the optimal discriminator in Eq.(10) can be formulated as:

$$D^*(x) = \frac{\sum_{P_\varepsilon \in S} P_{R,\varepsilon}(x)}{\sum_{P_\varepsilon \in S}[P_{R,\varepsilon}(x) + P_{G,\varepsilon}(x)]}. \tag{11}$$

Then, according to [22], the optimization of $G$ under optimal discriminator $D^*(x)$ for Eq.(10) can be formulated as:

$$\min_G \mathbf{JSD}(\frac{1}{|S|} \sum_{P_\varepsilon \in S} P_{R,\varepsilon} \,||\, \frac{1}{|S|} \sum_{P_\varepsilon \in S} P_{G,\varepsilon}), \tag{12}$$

where $\mathbf{JSD}$ is the Jensen-Shannon divergence. Following Theorem 1 in [22], we can simplify Eq.(12) as:

$$\min_G \mathbf{JSD}(\frac{1}{2}(P_R + P_R * P_\varepsilon) \,||\, \frac{1}{2}(P_G + P_G * P_\varepsilon)), \tag{13}$$

where $P_\varepsilon$ is a zero-mean Gaussian function produced by DANI with a positive definite covariance $\Sigma$. Then, based on the proofs of Theorem 1 in [22], we can conclude that the optimization of Eq.(13) is equal to the optimization between the distributions $P_G$ and $P_R$, which is the same as in the original GAN [14], indicating that the proposed DANI does not influence the convergence.

Next, we theoretically analyze the widely-used GANs with limited data backbones, i.e., StyleGAN2 and FastGAN with non-saturating loss and hinge loss, respectively.

### 3.2.1 Discussion of the non-saturating loss formulation in StyleGAN2. For the StyleGAN2, the non-saturating loss function can be formulated as:

$$V_D(D, G) = \mathbb{E}_{x \sim P_R}[\log D(x)] + \mathbb{E}_{x \sim P_G}[\log(1 - D(x))], \\ V_G(D, G) = -\mathbb{E}_{x \sim P_G}[\log(D(x))]. \tag{14}$$

Based on the theoretical analysis of the saturating loss above, the non-saturating loss with DANI can be formulated as:

$$V_D(D, G) = \sum_{P_\varepsilon \in S} \{\mathbb{E}_{\varepsilon \sim P_\varepsilon}[\mathbb{E}_{x \sim P_R}[\log D(x + \varepsilon)]] \\ + \mathbb{E}_{\varepsilon \sim P_\varepsilon}[\mathbb{E}_{x \sim P_G}[\log(1 - D(x + \varepsilon))]]\}, \tag{15}$$

$$V_G(D, G) = - \sum_{P_\varepsilon \in S} \mathbb{E}_{\varepsilon \sim P_\varepsilon}[\mathbb{E}_{x \sim P_G}[\log(D(x + \varepsilon))]].$$

Then, according to the original GAN and Theorem 2.5 as in [2], optimizing the non-saturating loss in GANs is equivalent to minimizing the **KL-2JS** divergence item (under optimal $D^*$). In this case, we follow [22] to consider images of $m \times n$ pixels and with values in a compact domain $\Omega \subset \mathbb{R}^{m \times n}$, then $P_R$ can be represented as $L^2(\Omega)$. In this case, the optimization of $G$ under Eq.(15) can be formulated as Theorem 1 as follows.

**Theorem 1.** Let us choose $S$ such that the optimization of $G$ in Eq.(15) can be written as

$$\min_G \mathbf{KLD}((P_G + P_G * P_\varepsilon) \,||\, (P_R + P_R * P_\varepsilon)) \\ - \mathbf{JSD}(\frac{1}{2}(P_R + P_R * P_\varepsilon) \,||\, \frac{1}{2}(P_G + P_G * P_\varepsilon)), \quad (16)$$

where $P_\varepsilon$ is a zero-mean Gaussian function produced by DANI with a positive definite covariance $\Sigma$. Let us also assume that the domain of $P_G$ is restricted to $\Omega$ (and thus $P_G \in L^2(\Omega)$). Then, the global optimum of Eq.(15) is $P_G(x) = P_R(x), \forall x \in \Omega$.
*Proof.* See supplementary materials.

#### 3.2.2 Discussion of the hinge loss formulation in FastGAN.
For the FastGAN, the hinge loss function can be formulated as:

$$V_D(D, G) = \mathbb{E}_{x \sim P_R}[\min(0, -1 + D(x))] \\ + \mathbb{E}_{x \sim P_G}[\min(0, -1 - D(x))], \quad (17) \\ V_G(D, G) = -\mathbb{E}_{x \sim P_G}[D(x)].$$

Based on the theoretical analysis of the saturating loss above, the hinge loss with DANI can be formulated as:

$$V_D(D, G) = \sum_{P_\varepsilon \in S} \{\mathbb{E}_{\varepsilon \sim P_\varepsilon}[\mathbb{E}_{x \sim P_R}[\min(0, -1 + D(x + \varepsilon))]] \\ + \mathbb{E}_{\varepsilon \sim P_\varepsilon}[\mathbb{E}_{x \sim P_G}[\min(0, -1 - D(x + \varepsilon))]]\}, \quad (18) \\ V_G(D, G) = -\sum_{P_\varepsilon \in S} \mathbb{E}_{\varepsilon \sim P_\varepsilon}[\mathbb{E}_{x \sim P_G}[D(x + \varepsilon)]].$$

According to [32, 38, 49], optimizing the hinge loss is equivalent to minimizing the so-called reverse **KL** divergence item (under optimal $D^*$). Therefore, following the definition in §3.2.1, the optimization of $G$ under Eq.(18) can be formulated as Theorem 2 as follows.
**Theorem 2.** Let us choose $S$ such that the optimization of $G$ in Eq.(18) can be written as

$$\min_G \mathbf{KLD}((P_G + P_G * P_\varepsilon) \,||\, (P_R + P_R * P_\varepsilon)), \quad (19)$$

where $P_\varepsilon$ is a zero-mean Gaussian function produced by DANI with a positive definite covariance $\Sigma$. Let us also assume that the domain of $P_G$ is restricted to $\Omega$ (and thus $P_G \in L^2(\Omega)$). Then, the global optimum of Eq.(18) is $P_G(x) = P_R(x), \forall x \in \Omega$.
*Proof.* See supplementary materials.

#### 3.2.3 Discussion of Regularization in StyleGAN2 and Fast-GAN.
Both StyleGAN2 and FastGAN apply regularization to $D$ in their loss function to enhance the training of GANs. To demonstrate the rigour and reasonableness of our theory above, we point out that regularization in both StyleGAN2 and FastGAN aims to avoid $D$ becoming overly confident. Therefore, regularization does not influence the soundness of our theoretical insights under optimal

$D^*$. Additionally, based on the theory in [37], applying regularization in GANs can still be convergent when initialized sufficiently close to the equilibrium point, which demonstrates that our theory analysis is reasonable.

Next, we prove the proposed DANI can increase the overlapping parts between the supports of $P_G$ and $P_R$, as shown in Theorem 3.
**Theorem 3.** For two supports of the distributions $P$ and $Q$ consisting of overlapping parts, if there exists one sample $x \in P \cap Q$, $\forall$ function $A_t$, $[x + pA_t(x)] \in [P + pA_t(P)] \cap [Q + pA(Q)]$.
*Proof.* $[x \in P \cap Q] \Rightarrow [x \in P$ and $x \in Q] \Rightarrow [pA_t(x) \in pA_t(P)$ and $pA_t(x) \in pA_t(Q)] \Rightarrow [x + pA_t(x) \in [P + pA_t(P)] \cap [Q + pA_t(Q)]]$.

The $P$ and $Q$ in Theorem 3 can be replaced as $P_G$ and $P_R$ for the GANs, and the function $A_t$ is the same as the adaptive forward diffusion function $A_t$ in DANI. Theorem 3 shows that with the function $A_t$, if the sample $x$ is in the overlapping parts between the supports of $P_G$ and $P_R$, $x + pA_t(x)$ can still be in the overlapping parts between the supports of $P_G + pA_t(P_G)$ and $P_R + pA_t(P_R)$. Because the probability $p$ and the function $A_t$ in DANI are both controlled by the overfitting degree of $D$ adaptively, this means that $p$ and $A_t$ will vary adaptively during training. In this case, one sample $x$ can produce a set of samples $\{x + pA_t(x)\}$ during training. This demonstrates that the overlapping parts between the DANI distribution $P_G + pA_t(P_G)$ and $P_R + pA_t(P_R)$ consist of more samples, thus providing more overlapping parts, leading to better performance.

### 3.3 Discussion of the Difference between DANI and Diffusion-GAN

Although the adaptive forward diffusion process has been already applied to the Diffusion-GAN, the two main differences between DANI and Diffusion-GAN are shown as follows. First, same to the data augmentation methods in GANs with limited data [26, 50, 54, 65], the noise injection formulation in Diffusion-GAN is to transform both real and fake samples by an adaptive forward diffusion process. This design can cause the original real and fake data not to be visible to $D$ during training, which decreases the training of GANs under limited data settings, leading to sub-optimal performance. In contrast, the noise injection form of the proposed DANI allows the original real and fake data to be visible to $D$ during training (under adaptive injection probability $p = 0$), which can improve the training of GANs under limited data settings, resulting in better performance. Second, according to [64], the leaking problem still exists in Diffusion-GAN. Compared with the Diffusion-GAN, which only employs the adaptive noise strength to alleviate the leaking problem, the proposed DANI has an additional adaptive injection probability to further alleviate the leaking problem, therefore leading to better performance.

## 4 EXPERIMENT

### 4.1 Datasets and Implementation Details

We select FFHQ [28], LSUN-CAT [46] and low-shot datasets for experiments. For fair comparisons, we follow the official open-source codes[1] for preprocessing and resizing the FFHQ and LSUN-CAT dataset to 256×256, as was the case in recent studies [26, 31, 65].

---

[1]https://github.com/NVlabs/stylegan2-ada-pytorch

**FID 10.08 (-1.13)**  **FID 3.04 (-0.94)**  **FID 14.92 (-0.88)**  **FID 17.72 (-0.29)**  **FID 16.81 (-1.07)**

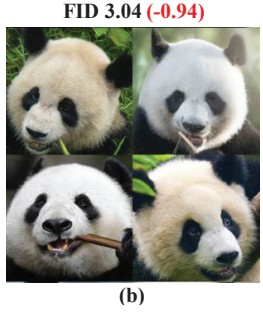 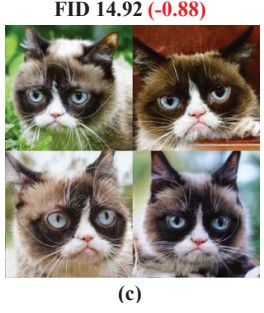 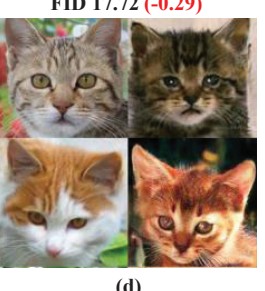 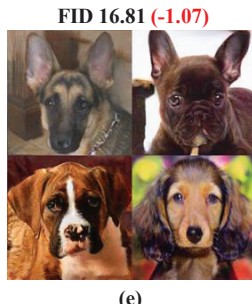

(a)  (b)  (c)  (d)  (e)

Figure 3: Images generated by Projected GAN + DANI on (a) 100-shot Obama dataset, (b) 100-shot Panda dataset, (c) 100-shot Grumpy Cat dataset, (d) Animal-Face Cat dataset and (e) Animal-Face Dog datasets. The decreasing value of FID in red color demonstrates the improvement of Projected GAN + DANI compared with baseline Projected GAN. *Best viewed in color.*

| Method | MA | Backbone | 100-shot | | | Animal-Face | |
| | | | Obama | Grumpy Cat | Panda | Cat | Dog |
|---|---|---|---|---|---|---|---|
| StyleGAN2 [28] | Yes | StyleGAN2 | 65.57 | 39.92 | 22.08 | 51.66 | 77.96 |
| Diff-Augment [65] | Yes | StyleGAN2 | 46.87 | 27.08 | 12.06 | 42.44 | 58.85 |
| ADA [26] | Yes | StyleGAN2 | 45.69 | 26.62 | 12.90 | 40.77 | 56.83 |
| Diffusion-GAN [54] | Yes | StyleGAN2 | 28.55 | 21.87 | 8.69 | 33.18 | 68.15 |
| AugSelf-StyleGAN2 [21] | Yes | StyleGAN2 | 26.00 | 19.81 | 8.36 | 30.53 | 48.19 |
| FakeCLR [31] | Yes | StyleGAN2 | 26.95 | 19.56 | 8.42 | 26.34 | 42.02 |
| InsGen [57] | Yes | StyleGAN2 | 32.42 | 22.01 | 9.85 | 33.01 | 44.93 |
| **InsGen + DANI** | Yes | StyleGAN2 | **23.25** | **18.83** | **6.99** | **24.14** | **34.19** |
| FastGAN [33] | Yes | FastGAN | 35.80 | 25.75 | 9.70 | 33.85 | 52.46 |
| FreGAN [58] | Yes | FastGAN | 33.39 | 24.93 | 8.97 | 31.05 | 47.85 |
| Projected GAN [45] | Yes | FastGAN | 11.21 | 15.80 | 3.98 | 18.01 | 17.88 |
| **Projected GAN + DANI** | Yes | FastGAN | **10.08** | **14.92** | **3.04** | **17.72** | **16.81** |

Table 1: FID score (lower is better) on several low-shot datasets ($256 \times 256$). We follow the setting as in [65]. MA means Massive Augmentation, i.e., xflipping, which has the same meaing as in [9]. For a fair comparison, the FIDs are averaged over five runs; all standard deviations are less than 1% relatively. The results of the Projected GAN are run by ourselves based on the official open-source codes https://github.com/autonomousvision/projected-gan.

| Method | MA | Backbone | FFHQ | | | |
| | | | 100 | 1K | 2K | 5K |
|---|---|---|---|---|---|---|
| StyleGAN2 [28] | Yes | StyleGAN2 | 179 | 100.16 | 54.3 | 49.68 |
| ADA [26] | Yes | StyleGAN2 | 85.8 | 21.29 | 15.39 | 10.96 |
| Diff-Augment [65] | Yes | StyleGAN2 | 61.91 | 25.66 | 24.32 | 10.45 |
| APA [23] | Yes | StyleGAN2 | 65 | 18.89 | 16.90 | 8.38 |
| AugSelf-StyleGAN2+ [21] | Yes | StyleGAN2 | - | 20.39 | - | 9.15 |
| FakeCLR [31] | Yes | StyleGAN2 | 42.56 | 15.92 | 9.90 | 7.25 |
| InsGen [57] | Yes | StyleGAN2 | 45.75 | 18.21 | 11.47 | 7.83 |
| **InsGen + DANI** | Yes | StyleGAN2 | **41.79** | **15.63** | **9.78** | **7.21** |
| Projected GAN | Yes | FastGAN | 26.25 | 11.12 | 8.25 | 6.85 |
| **Projected GAN + DANI** | Yes | FastGAN | **23.98** | **10.81** | **7.73** | **6.20** |

Table 2: FID score (lower is better) on the $256 \times 256$ FFHQ dataset. Following FakeCLR [31], we perform experiments on 100, 1K, 2K and 5K training samples on the FFHQ dataset. MA means Massive Augmentation, i.e., xflipping, which has the same meaing as in [9]. For a fair comparison, the FIDs are averaged over five runs; all standard deviations are less than 1% relatively. The results of the Projected GAN are run by ourselves based on the official codes https://github.com/autonomousvision/projected-gan.

| Method | MA | Backbone | LSUN-CAT | | | |
|---|---|---|---|---|---|---|
| | | | 1K | 5K | 10K | 30K |
| Projected GAN | Yes | FastGAN | 19.39 | 13.57 | 8.92 | 8.65 |
| **Projected GAN + DANI** | Yes | FastGAN | **16.82** | **11.27** | **8.53** | **8.21** |

Table 3: FID score (lower is better) on the $256 \times 256$ LSUN-CAT dataset. Following Diff-Augment [65], we perform experiments on 1K, 5K, 10K and 30K training samples on the LSUN-CAT dataset. MA means Massive Augmentation, i.e., xflipping, which has the same meaing as in [9]. For a fair comparison, the FIDs are averaged over five runs; all standard deviations are less than 1% relatively. The results of the Projected GAN are run by ourselves based on the official open-source codes.

The FFHQ dataset contains 70K high-resolution images of human faces. Following FakeCLR [31], we select a subset of 100, 1K, 2K, and 5K for a fair comparison. The LSUN-CAT dataset contains 200K high-resolution images of cats. Based on Diff-Augment [65], we select a subset of 1K, 5K, 10K and 30K for the comparison. The Low-shot datasets contain five datasets (Obama, Grumpy Cat, Panda, Animal-Face Cat, and Animal-Face Dog) with 100, 100, 100, 160, and 389 training images, respectively. We select the state-of-the-art StyleGAN2 and FastGAN methods, e.g., InsGen and Projected GAN [45], as our backbone and set the batch size as 64 for all experiments. More importantly, according to Eq.(2), the DANI in Projected GAN is applied to the real and fake projected features rather than real and fake images. The commonly used Fréchet Inception Distance (FID) [19] is applied as the evaluation metric; the full dataset is used as the reference distribution for FID calculation, following prior work [26, 31, 65]. For the adaptive forward diffusion process $A_t$ in DANI, based on Diffusion-GAN [54], the initial value of $t$ is set as 5. Then, we set $t_{min} = 5$ and the $t_{max} = 1000$ for InsGen, and set the $t_{min} = 5$ and the $t_{max} = 500$ for Projected GAN. The forward diffusion process variances are set to constants increasing linearly from $\beta_1 = 10^{-4}$ to $\beta_T = 0.02$. More details of experiments, i.e., experimental results with the other evaluation metric Inception Score (IS) [44] and the link of the pre-trained model with test code, can be found in supplementary materials.

## 4.2 Results on Low-shot Datasets

The results on low-shot datasets with StyleGAN2 and FastGAN are shown in Table 1. Applying DANI to both InsGen and Projected GAN can achieve better performance. Projected GAN + DANI achieves the lowest FID compared with all of the other methods. More importantly, Projected GAN has already applied DA, regularizations and pre-trained models to achieve state-of-the-art performance on the low-shot datasets; adding DANI can further reduce the FID score by about 5% to 10% and achieve novel state-of-the-art performance. The images generated by Projected GAN + DANI on low-shot datasets are shown in Figure 3.

## 4.3 Results on FFHQ and LSUN-CAT Datasets

The results on the FFHQ datasets are shown in Table 2. For the FFHQ dataset, following the FakeCLR [31], we perform experiments on the subset of 100, 1K, 2K and 5K. Applying DANI to both InsGen and Projected GAN can achieve better performance and Projected GAN + DANI achieves state-of-the-art performance compared with other methods. The results on the LSUN-CAT datasets with the state-of-the-art method, i.e., Projected GAN, are shown in Table 3.

For the LSUN-CAT dataset, based on Diff-Augment [65], we conduct experiments on the subset of 1K, 5K, 10K and 30K. Applying DANI to Projected GAN also achieves better performance.

## 4.4 Computational Cost

The results of the training time on the Animal-Face Dog dataset ($256 \times 256$) with or without DANI using Projected GAN have been demonstrated in Table 4. The increased computational cost with DANI is negligible (< 1%).

| Method | Seconds per 1K images |
|---|---|
| Projected GAN | 12.11 |
| **Projected GAN + DANI** | 12.20 |

Table 4: The training time on the Animal-Face Dog dataset ($256 \times 256$) with or without DANI. The results are calculated by averaging over ten times on the four NVIDIA RTX TITAN GPUs with batch size 64. All standard deviations are less than 1% relatively.

## 4.5 Ablation Study

**Impact of two adaptive strategies in DANI.** To demonstrate that two adaptive strategies in DANI are reasonable, we conduct an ablation study on the impact of adaptive strategies in DANI, and the results are shown in Table 6. It is clear that directly adding the Gaussian noise to the Projected GAN, i.e., Projected GAN + DANI (without both adaptive strategies), can cause worse performance compared with the baseline. Adding each adaptive strategy in DANI to the Projected GAN increases the performance, which demonstrates that the two adaptive strategies in DANI are suitable. Furthermore, to further demonstrate that DANI can alleviate the leaking problem, following [64], we also show the compared generated images of Projected GAN + DANI (without both adaptive strategies) and Projected GAN + DANI on the 100-shot Obama dataset, as shown in Figure 5. It is clear that directly adding the Gaussian noise to the Projected GAN, i.e., Projected GAN + DANI (without both adaptive strategies), still has slightly noisy results. In contrast, applying DANI to Projected GANs, i.e., Projected GAN + DANI, can avoid the leaking issue during training.

**DANI v.s. Transforming both real and fake data by an adaptive forward diffusion.** Recently, Diffusion-GAN [54] transforms both real and fake samples by an adaptive and forward diffusion to improve GANs training. To further show the superiority of DANI compared with Diffusion-GAN, we also conduct an ablation study

FID 23.98 **(-2.27)**      FID 10.81 **(-0.31)**      FID 7.73 **(-0.52)**      FID 6.20 **(-0.65)**

(a)        (b)        (c)        (d)

**Figure 4: Images generated by Projected GAN + DANI on (a) FFHQ-100 dataset, (b) FFHQ-1K dataset, (c) FFHQ-2K dataset and (d) FFHQ-5K dataset. The decreasing value of FID in red color demonstrates the improvement of Projected GAN + DANI compared with baseline Projected GAN.** *Best viewed in color.*

| Method | MA | 100-shot | | | Animal-Face | |
|---|---|---|---|---|---|---|
| | | Obama | Grumpy Cat | Panda | Cat | Dog |
| Diffusion-Projected GAN [54] | Yes | 10.54 | 15.13 | 3.39 | 17.86 | 17.22 |
| **Projected GAN + DANI** | Yes | **10.08** | **14.92** | **3.04** | **17.72** | **16.81** |

**Table 5: FID score (lower is better) on several low-shot datasets ($256 \times 256$). We follow the setting as in [65]. MA means Massive Augmentation, i.e., xflipping, which has the same meaning as in [9]. For a fair comparison, the FIDs are averaged over three runs; all standard deviations are less than 1% relatively. The results of the Diffusion-Projected GAN are run by ourselves based on the official open-source codes.**

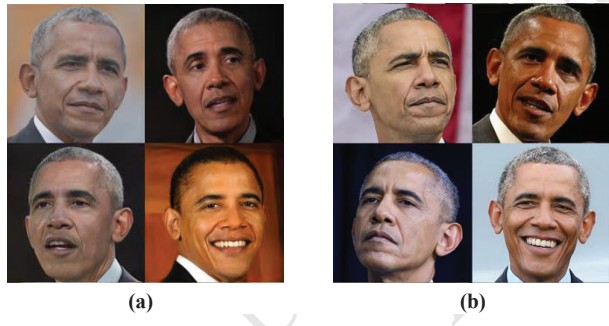

(a)           (b)

**Figure 5: Compared generated images on the 100-shot Obama dataset: (a) Images generated by Projected GAN + DANI (without both adaptive strategies) and (b) Images generated by Projected GAN + DANI.** *Best viewed in color.*

| Method | FID |
|---|---|
| Projected GAN (baseline) | 11.21 |
| Projected GAN + DANI (*w/o* both strategies) | 11.62 |
| Projected GAN + DANI (*w/o* adaptive noise strength) | 10.81 |
| Projected GAN + DANI (*w/o* adaptive injection probability) | 10.33 |
| **Projected GAN + DANI** | **10.08** |

**Table 6: FID score (lower is better) on the 100-shot-Obama dataset ($256 \times 256$). Massive Augmentation [9] is applied to all of the methods. For a fair comparison, the FIDs are averaged over three runs; all standard deviations are less than 1% relatively.**

comparing the proposed DANI with Diffusion-GAN with the Projected GAN backbone, and the results are shown in Table 5. Projected GAN + DANI can achieve better performance compared with Diffusion-Projected GAN on low-shot datasets. More comparison results on the FFHQ dataset can be found in supplementary materials.

## 5 CONCLUSION

In this paper, to improve GANs training with limited data, we propose a novel noise injection method called Dual Adaptive Noise Injection (DANI), with a negligible computational cost increase. Specifically, DANI consists of two adaptive strategies, i.e., the adaptive injection probability and adaptive noise strength. For the adaptive injection probability, we inject Gaussian noise into both real and fake images for $G$ and $D$ through a probability $p$, where $p$ is controlled adaptively by the overfitting degree of $D$. For adaptive noise strength, the Gaussian noise is produced by applying the adaptive forward diffusion process to the images. Extensive experiments on several commonly-used datasets demonstrate that DANI can effectively improve the training of GANs with limited data and achieve state-of-the-art results compared with other methods.

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
