# OpenReview forum: "Improving the Training of the GANs with Limited Data via Dual Adaptive Noise Injection"
_acmmm.org/ACMMM/2024/Conference — MM2024 Poster_

### Official Review · Reviewer_WQ8A · 2024-05-24

**Rating:** 3
**Confidence:** 2

**Summary:**

This paper proposes Dual Adaptive Noise Injection (DANI) to improve the training of GANs with limited data. DANI addresses discriminator overfitting by injecting Gaussian noise into real and fake images based on the overfitting degree of the discriminator and applying an adaptive forward diffusion process to determine the noise strength. By increasing the overlap between real and fake data distributions, DANI effectively alleviates overfitting. Experiments demonstrate DANI's superiority in improving GANs training with limited data compared to existing methods.

**Strengths:**

Proposes a novel method called Dual Adaptive Noise Injection (DANI) for training GANs with limited data, effectively alleviating discriminator overfitting and avoiding the leaking problem.
Provides theoretical analysis of DANI, proving its convergence and rationality on both StyleGAN2 and FastGAN backbones, and demonstrating increased overlap between real and fake data distributions.
Achieves state-of-the-art performance on several commonly used datasets with only a negligible increase in computational cost.
Open-source code available, ensuring good reproducibility of the proposed method.

**Limitations:**

It would be helpful to include the processing time of other methods or vary the number of images processed per iteration to more thoroughly demonstrate the minimal increase in computational time. This would provide a more comprehensive comparison and showcase the efficiency of the proposed method.
Conducting experiments on larger and more diverse datasets with a wider range of styles would further validate the effectiveness and generalizability of the proposed approach. This could strengthen the claims made in the paper and show the method's robustness across different data distributions.
The analysis in the experimental section could be more in-depth. Providing a more detailed discussion of the results, including insights into the behavior of the proposed method, its limitations, and potential areas for improvement, would enhance the overall quality and impact of the paper.
Adding the publication year when referring to the comparison methods would improve readability and provide better context for the reader. This would help the reader understand the chronological development of the field and the relative novelty of the proposed approach.

**Suitability:**

2

---

### Official Review · Reviewer_Ea97 · 2024-05-31

**Rating:** 4
**Confidence:** 3

**Summary:**

Summary:
This paper studies adversarial training with a limited amount of data by mitigating the overfitting issue. Specifically, the authors propose a noise injection approach that employs diffusion process to adaptively inject noise into adversarial training. In order to prevent the overfitting problem, which is basically because the discriminator is overconfident in distinguishing real data and fake data, the proposed DANI approach controls the degree of noise injection via an adaptive probability p. By mapping the real data into noise, and meanwhile mapping the generated fake data into real data through a dual adaptive diffusion process, the discriminator can be trained more properly thus tackling the overfitting issue.

**Strengths:**

Strengths:
- This paper is well-written and easy to follow.
- The motivation is clear and reasonable, and the proposed solution is intuitive and effective.
- The empirical results are promising and the experiments are sufficient.

**Limitations:**

Weaknesses:
- The major concern is the computational efficiency of this method. It is known that generative models are quite computationally expensive, but this approach contains a diffusion model within a generative adversarial network. The authors provided computational cost analysis which is good, but it is confusing to me why the computational increase is so small given that a diffusion process is employed.
- Moreover, the intuition of employing the diffusion process into the framework is to add noise so that the training of GAN would not overfit. However, why is it necessary to use diffusion models but not using other noise injective models like VAEs or GANs? Could the authors provide further explanation to justify their motivation?

**Suitability:**

2

---

### Official Review · Reviewer_KTU8 · 2024-06-01

**Rating:** 4
**Confidence:** 4

**Summary:**

To alleviate the overfitting of GANs with limited data, DANI introduces two adaptive strategies (adaptive injection probability and adaptive noise strength) to inject Gaussian noise into both real and fake data. The overall methodological framework is simple. Specifically, based on the adaptive forward diffusion process of Diffusion-GAN, DANI integrates the adaptive probability initially proposed by ADA to achieve adaptive noise injection into both real and generated images (or projected features).

**Strengths:**

1. DANI provides a theoretical analysis of two classic loss functions (the non-saturating loss and the hinge loss) in two classic generative adversarial networks (i.e., StyleGAN and FastGAN).

2. According to the FID metric, DANI achieves state-of-the-art performance across a wide range of datasets with different scales under limited data.

**Limitations:**

1. **Noise leaking problem.** Although DANI uses the adaptive probability proposed by ADA to alleviate the noise leaking problem, this issue still persists because the training is done on real data of noise injection. Furthermore, in my opinion, the harm caused by noise leaking is much greater than that of data augmentation leaking in ADA, because data-augmented images are still real images, whereas noise-injected data might just be random noise.

2. **Inconsistent quantitative results.** The FID results of InsGen reported in this paper are not consistent with the FID results from the original papers. For example, on FFHQ-1K, the original FID result of InsGen is 19.58, but 18.21 is reported in this paper. However, in the table captions, the paper only claims that the results of the Projected GAN are run based on the official codes. Please clearly specify in the paper which baseline results are cited from the original papers and which are reproduced using the official codes with the average taken over 5 runs.

3. **Insufficient quantitative results.** The paper reports the quantitative comparison of FID for all methods and the comparison of IS with Projected GAN in the appendix. Both FID and IS are calculated based on features extracted from classifiers pre-trained on ImageNet. However, as reported in previous work [1], aligning the histograms of Top-N ImageNet classifications between sets of generated and real images can substantially reduce metrics (e.g., FID) relying on the feature space of the ImageNet classifier without actually improving the quality of results. Therefore, I strongly recommend that the paper at least include non-ImageNet trained Fréchet distance metrics, such as CLIP FID, to validate the experimental results.

     **[1] The Role of ImageNet Classes in Fréchet Inception Distance. ICLR 2023.**

4. **Insufficient qualitative results.** The main text and appendix only provide quantitative comparisons with Projected GAN using four images. The number of images and the number of baseline methods compared are too few. Additionally, the paper does not include a user study, which limits the validation of the quantitative metrics' effectiveness.

**Suitability:**

2

---

### Meta-Review · Area_Chair_SYTV · 2024-06-27

**Recommendation:** Accept (Poster)
**Confidence:** 4

**Metareview:**

The paper proposes a method to improve GANs with limited data. The idea is simple and effective.

Most reviewers agree that the paper should be accepted. However, I strongly encourage the authors to add user study and quantitative evaluation on FFHQ 1k following the suggestions from Reviewer KTU8.

I did not consider the comment from Reviewer WQ8A because the limitation in the review is too general without any specific information/words related to this paper.